# Acute febrile illness in Kenya: Clinical characteristics and pathogens detected among patients hospitalized with fever, 2017–2019

Jennifer R. Verani[1,2]*, Eric Ng' eno[3], Elizabeth A. Hunsperger[1,2], Peninah Munyua[2], Eric Osoro[3], Doris Marwanga[4], Godfrey Bigogo[4], Derrick Amon[4], Melvin Ochieng[4], Paul Etau[5], Victor Bandika[6], Victor Zimbulu[7], John Kiogora[8], John Wagacha Burton[9], Emmanuel Okunga[10], Aaron M. Samuels[11], Kariuki Njenga[3], Joel M. Montgomery[1], Marc-Alain Widdowson[1,2]

1 Division of Global Health Protection, U.S. Centers for Disease Control and Prevention, Atlanta, GA, United States of America, 2 Division of Global Health Protection, U.S. Centers for Disease Control and Prevention, Nairobi, Kenya, 3 Washington State University Global Health, Nairobi, Kenya, 4 Centre for Global Health Research, Kenya Medical Research Institute, Kisumu, Kenya, 5 Kenyatta National Hospital, Nairobi, Kenya, 6 Coast General Provincial Hospital, Mombasa, Kenya, 7 Kakamega County Referral Hospital, Kakamega, Kenya, 8 International Rescue Committee, Nairobi, Kenya, 9 United Nations High Commissioner for Refugees, Nairobi, Kenya, 10 Disease Surveillance and Response Unit, Ministry of Health, Nairobi, Kenya, 11 Division of Parasitic Diseases and Malaria, Center for Global Health, Centers for Disease Control and Prevention, Kisumu, Kenya and Atlanta, Georgia, United States of America

* jverani@cdc.gov

**Data Availability Statement:** Data cannot be shared publicly because they are bound by Government of Kenya provisions, including the Data Protection Act of 2019. Data are available

## Abstract

Acute febrile illness (AFI) is a common reason for healthcare seeking and hospitalization in Sub-Saharan Africa and is often presumed to be malaria. However, a broad range of pathogens cause fever, and more comprehensive data on AFI etiology can improve clinical management, prevent unnecessary prescriptions, and guide public health interventions. We conducted surveillance for AFI (temperature ≥38.0˚C <14 days duration) among hospitalized patients of all ages at four sites in Kenya (Nairobi, Mombasa, Kakamega, and Kakuma). For cases of undifferentiated fever (UF), defined as AFI without diarrhea (≥3 loose stools in 24 hours) or lower respiratory tract symptoms (cough/difficulty breathing plus oxygen saturation <90% or [in children <5 years] chest indrawing), we tested venous blood with real-time PCR-based TaqMan array cards (TAC) for 17 viral, 8 bacterial, and 3 protozoal fever-causing pathogens. From June 2017 to March 2019, we enrolled 3,232 AFI cases; 2,529 (78.2%) were aged <5 years. Among 3,021 with outcome data, 131 (4.3%) cases died while in hospital, including 106/2,369 (4.5%) among those <5 years. Among 1,735 (53.7%) UF cases, blood was collected from 1,340 (77.2%) of which 1,314 (98.1%) were tested by TAC; 715 (54.4%) had no pathogens detected, including 147/196 (75.0%) of those aged <12 months. The most common pathogen detected was *Plasmodium*, as a single pathogen in 471 (35.8%) cases and in combination with other pathogens in 38 (2.9%). HIV was detected in 51 (3.8%) UF cases tested by TAC and was most common in adults (25/236 [10.6%] ages 18–49, 4/40 [10.0%] ages ≥50 years). Chikungunya virus was found in 30 (2.3%) UF cases, detected only in the Mombasa site. Malaria prevention and control efforts are critical for

from KEMRI via the Data Governance Committee (contact via email cghr@kemri.go.ke or telephone +254-20-22923) to researchers who meet the criteria for access to confidential data and with permission of Kenya Ministry of Health.

**Funding:** This work was supported by an award from the Centers for Disease Control and Prevention (CDC) to Washington State University for the implementation of the surveillance program (Grant no. 1U01GH002143). The authors affiliated with Washington State University, namely Eric Ng'eno, Eric Osoro, and Kariuki Njenga, acknowledge the specific funding received for this research.

**Competing interests:** The authors have declared that no competing interests exist.

reducing the burden of AFI, and improved diagnostic testing is needed to provide better insight into non-malarial causes of fever. The high case fatality of AFI underscores the need to optimize diagnosis and appropriate management of AFI to the local epidemiology.

## Introduction

Acute febrile illness (AFI) is a common reason for healthcare seeking and hospitalization in Sub-Saharan Africa, with more than 16 million hospital admissions for severe febrile illness estimated per year in the region [1, 2]. Fever is often an early presenting sign of the leading infectious causes of morbidity and mortality in Sub-Saharan Africa, including, pneumonia, diarrheal diseases, malaria, HIV/AIDS, and tuberculosis [3]. Fever is also a prominent clinical feature of several outbreak-prone diseases, including arboviruses (e.g. dengue and chikungunya), viral hemorrhagic fevers (e.g. Rift Valley fever, Ebola, Marburg, and Lassa), typhoid fever, and respiratory viruses (e.g., influenza, respiratory syncytial virus, and coronaviruses).

Despite this high burden, the etiology of most AFI cases in Sub-Saharan Africa remains unknown [4, 5]. Suboptimal diagnostic capabilities at health facilities limits the range of pathogens that can be examined. Within malaria-endemic regions of Africa, fever is generally presumed to be indicative of malaria, and testing for other pathogens is frequently unavailable or often not performed [6]. One main limitation of studies of AFI has been that they often focus on a single pathogen or a specific age group, providing a narrow insight into AFI epidemiology [7]; additionally, pathogen distribution and associated burden is highly heterogeneous, limiting the generalizability of study results. Also, many studies were conducted for short time periods and may have failed to detect pathogens that have seasonal patterns.

Recent comprehensive studies of the burden and etiology of pneumonia [8] and diarrhea [9] in children provided important insight into the causes of those clinical syndromes in resource-poor settings. However, AFI as a clinical syndrome is less well understood, particularly fever in the absence of pneumonia or diarrhea. Understanding the causes of AFI in sub-Saharan African countries can strengthen clinical management, improve rational use of antibiotics, and guide the development of interventions and policy decisions for the prevention and control of AFI, and help understand the background risk of outbreak-prone pathogens.

We conducted surveillance for AFI among hospitalized patients of all ages in four ecologically distinct sites across Kenya to describe the clinical characteristics and etiology of patients hospitalized with fever and tested a subset of cases for a wide array of pathogens to better understand AFI etiologies in these contexts.

## Methods

AFI surveillance was conducted at 4 hospitals located in Nairobi, Mombasa, Kakamega, and Kakuma (S1 Fig). Kenyatta National Hospital is the largest tertiary national teaching and referral hospital in Kenya, with a capacity of 2000 beds. Situated in Nairobi, with a warm and temperate climate at 1600m altitude with little malaria or arboviruses, Kenyatta National Hospital offers emergency and inpatient services for patients referred from secondary and tertiary healthcare facilities in Nairobi and across the country. Coast General Teaching and Referral Hospital is a 700-bed facility located in Mombasa, the second largest city in Kenya, and in a setting with a tropical wet and dry climate. The facility serves both urban and rural populations from the coastal region of Kenya, a region affected by mosquito-borne viral diseases [10, 11] and with an intermediate burden of malaria [12]. The 500-bed Kakamega County Referral

Hospital is located in a predominantly rural, high malaria burden [12] western region of Kenya that experiences rainfall throughout the year. Ammusait General Hospital is a 200-bed facility located within the Kakuma refugee camp in Turkana County, an arid area of northwest Kenya. The facility is run by the International Rescue Commission (IRC) and mainly serves more than 190,000 persons displaced from more than 10 countries (predominantly from South Sudan and Somalia) [13] and the surrounding host communities who are pastoralists. The region has a high incidence of malnutrition among children [14], and outbreaks of malaria [15] and cholera [16] have occurred in the refugee population. The estimated county-level seroprevalence of HIV among adults aged 15–49 years in 2018 for the four surveillance sites was: 3.3% in Nairobi, 5.1% in Mombasa, 3.1% in Kakamega, and 7.1% in Turkana (Kakuma site) [17]. Implementation of the surveillance began in the Nairobi site on June 2, 2017, followed by Kakuma on August 22, 2017, and Mombasa and Kakamega on January 2, 2018. This analysis included data on patients enrolled at all sites through March 31, 2019.

At each site, trained surveillance officers reviewed admissions logs daily for pediatric and adult medical wards. All new admissions were screened for eligibility based on information in the medical record. Those eligible were approached for consenting and enrolment into the study. We enrolled all patients who met the AFI case definition: patients with a temperature ≥38.0˚C on admission, onset <14 days prior to presenting at the facility. Patients who were readmitted to the hospital within 14 days of having been previously enrolled and those primarily seeking care for injury or trauma (even if fever was present) were excluded. Among enrolled AFI cases, we identified those presenting with undifferentiated fever (UF), defined as AFI without diarrhea (≥3 loose stools in 24 hours) or lower respiratory tract infection (cough or difficulty breathing plus oxygen saturation <90% or [in children <5 years] indrawing).

Trained surveillance officers interviewed newly admitted patients (or parents/guardians of minors) using a standardized questionnaire to gather demographic, socioeconomic, clinical, and risk factor data, followed by a physical examination. After enrolled patients were discharged from the facility, medical charts were reviewed and data on clinical course, management, and outcome were abstracted. Blood was collected via venipuncture only from UF cases to test for potential AFI etiologies. The volume of blood drawn varied by age group and whether blood culture was performed.

All sites except Mombasa collected blood for culture. For blood culture, 1–3 mLs of whole blood for cases aged <10 years and 8–10 mLs for cases aged ≥10 years were inoculated into a commercially produced broth bottle (Peds Plus™/F BACTEC™ Plus and Aerobic/F culture vials, respectively, Becton Dickinson, Belgium). Inoculated blood culture bottles were incubated in a continuously monitored BACTEC instrument at 35˚C for up to 5 days. In Nairobi, blood culture was performed at a local Kenya Medical Research- Centre for Global Health Research (KEMRI-CGHR) laboratory supported by the U.S. Centers for Disease Control and Prevention (CDC) that was located ~8.5 km from Kenyatta National Hospital). In Kakuma, culture was performed in the Ammusait General Hospital laboratory. In Kakamega, inoculated bottles were incubated on site and transported to a CDC-supported KEMRI-CGHR laboratory in Kisumu (~43 km from the Kakamega site) only if there was evidence of growth indicated by BACTEC alarm. Samples from bottles with evidence of growth were Gram strained, plated on blood agar plate, chocolate agar plate, and MacConkey plates, and incubated aerobically and anaerobically at 37˚C for 24 hours. Identification was carried out through colony morphology, Gram stain and biochemical tests.

Whole blood was collected in EDTA tubes and stored at -20˚C at each surveillance facility for up to 7 days before being shipped for storage and testing at the KEMRI-CGHR laboratory in Nairobi. Whole blood was tested with a real-time PCR-based Taqman Array Card (TAC) designed for AFI surveillance that included 17 viral targets (chikungunya, Crimean-Congo

hemorrhagic fever, dengue [serotypes 1–4], Bundibugyo ebolavirus, Sudan ebolavirus, hepatitis E, Lassa, Marburg, Rift Valley fever, Nipah, West Nile, O'nyong-nyong, yellow fever, Zika, HIV I and II), 8 bacterial targets (*Brucella* spp., *Bartonella* spp., *Coxiella burnetii*, *Leptospira* spp., *Rickettsia* spp., *Salmonella enterica*, *Salmonella enterica* serovar Typhi, *Yersinia pestis*) and 3 protozoal targets (*Plasmodium* spp., *Leishmania* spp., *Trypanosoma brucei*) (S2 Fig). Methods for TAC testing have been previously described [18]. Briefly, nucleic acid was extracted from 2ml of whole blood (1 ml for children aged <5 years) using High Pure Extraction kit (Roche) and purified following established procedures. Approximately 46ul of purified nucleic acid was mixed with AgPath one step RT-PCR reagents (Thermo Fisher) and added to inlet portal of the TAC following manufacturer's instructions. The cards were run on Viia7 real time PCR system (ABI technologies) using cycling conditions of 10 minutes at 50°C, 20 seconds at 45°C, 10 minutes at 95°C followed by 45 two-step cycles of 15 seconds at 95°C and 1 minute at 60°C. The sample was designated positive when sample well reactions yielded amplification of Cycle Threshold (Ct) <37.

All analyses were done using STATA 12 (Stata Corporation, College Station, TX, USA) or SAS 9.4 (SAS, Cary, NC, USA). We summarized AFI and UF case demographic, socio-economic, clinical characteristics and outcomes using counts, proportions and charts for categorical variables and used means, median and interquartile range (IQR) for continuous variables. The frequencies of pathogens were presented as proportions and charts were generated by age group and by site.

This study was reviewed and approved by the Kenya Medical Research Institute's Scientific and Ethical Review Unit (number SSC 2980), the Institutional Review Board for US Centers for Disease Control and Prevention (protocol number 6757), site-specific ethical review boards at Kenyatta National Hospital and Coast General Teaching and Referral Hospital, and Washington State University Institutional Review Board (approval provided on basis of reliance on in-country ethical reviews). Administrative approval was provided by the Kenya Ministry of Health. Written informed consent was obtained from all participants. For children aged <7 years, parental consent was provided by the parent or guardian while for children aged 7–17 years, parental consent and additional written assent from the child was obtained.

## Results

From June 2, 2017 through March 31, 2019, 5,152 eligible AFI cases were identified at all four sites, of whom 3,232 (62.7%) were enrolled (Fig 1), including 951 in Kakuma, 396 in Kakamega, 1,140 in Nairobi, and 745 in Mombasa (Table 1). Among 1,920 (37.3%) not enrolled, the most common reasons for nonenrolment were declined participation (42.7%) and inability of the surveillance staff to locate the eligible patient (e.g., patient was already discharged or transferred) (37.7%). Among enrolled AFI cases, 1,735 (53.7%) met UF criteria; of those, 1,340 (77.2%) had blood collected. Across all sites enrolled participants were predominantly children; 2,529 (78.2%) AFI cases and 1,139 (65.6%) UF cases were aged <5 years (Table 1). Overall, 1,774 (54.9%) AFI cases and 931 (53.7%) UF cases were male.

Overall, the symptoms most frequently reported by AFI cases in addition to fever were cough (57.0%), vomiting (37.9%) and diarrhea (39.9%) (Table 1). Among UF cases, the most common symptoms reported were headache (47.7%), cough (47.6%) and lack of appetite (41.4%). On physical examination, the most frequent findings noted for AFI cases were lethargy (38.3%), tachypnea (31.0%) and rales/crackles (27.5%); the same three findings were most common among UF cases. The most common discharge diagnoses recorded for AFI cases were pneumonia (36.6%), malaria (21.2%) and gastroenteritis (19.7%); among UF cases malaria diagnosis was most common (31.5%), followed by pneumonia (23.4%) (S1 Table).

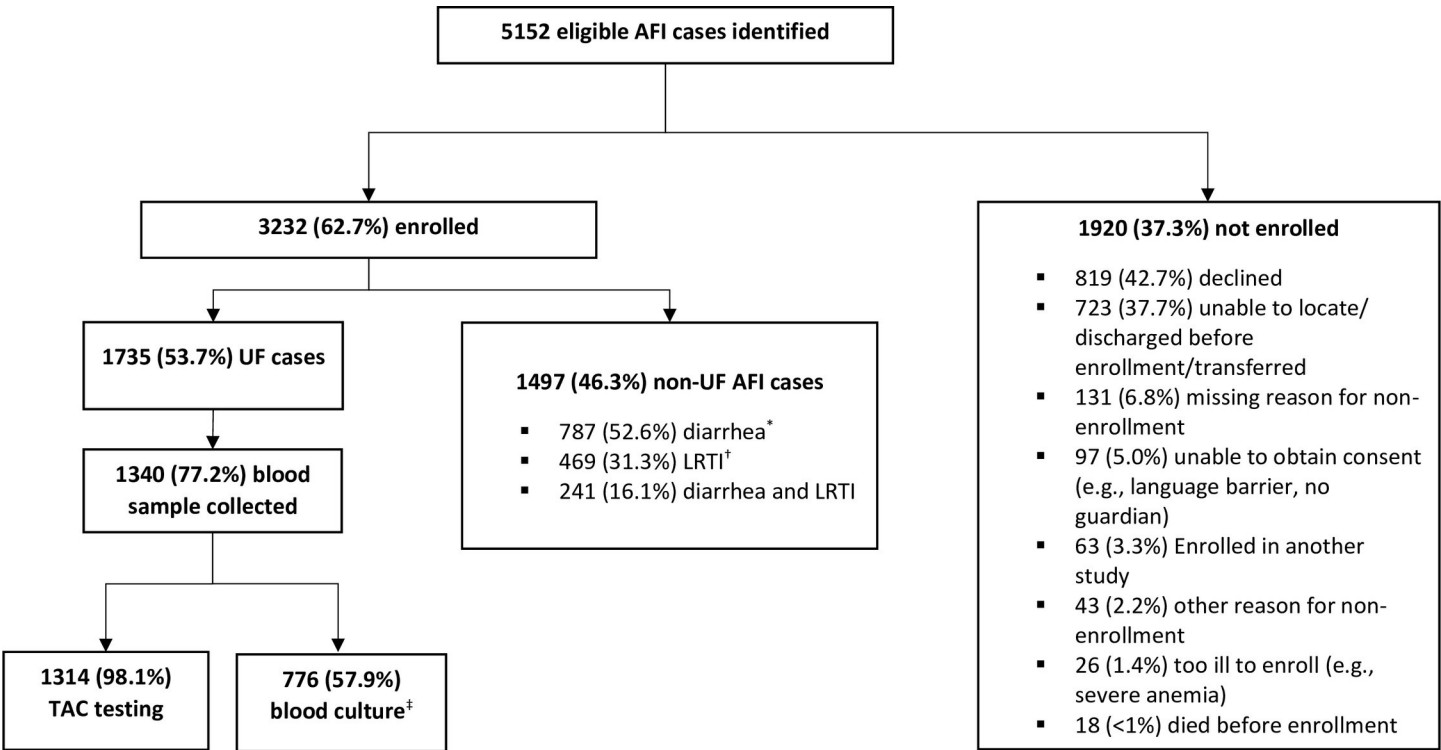

**Fig 1. Eligibility and enrollment of acute febrile illness (AFI) and undifferentiated fever (UF) cases, at four hospitals in Kenya, June 2017-March 2019.** *Diarrhea defined as ≥3 loose stools in 24-hour period. † LRTI = lower respiratory tract infection, defined as cough or difficulty breathing plus tachypnea (and/or chest-wall indrawing among patients aged <5 years). ‡ Blood culture performed at 3 of 4 sites.

Among 3,021 AFI cases with outcome data, 131 (4.3%) died while in the hospital. The in-hospital case fatality was 4.5% (106/2369) among AFI cases aged <5 years and 3.8% (25/652) among cases aged ≥5 years. By site, all-age AFI case fatality ranged from 1.7% in Kakuma to 7.3% in Nairobi.

The most commonly reported chronic medical condition overall was malnutrition (8.7%; by site ranging from 5.1% in Kakuma to 13.0% in Nairobi). The frequency of self-reported immunodeficiency, including HIV, was 2.6% overall and by site ranged from 1.1% in Kakuma to 3.8% in Nairobi. Hospitalization in the prior 12 months was reported by 19.3% of cases aged <5 years and 5.2% of cases aged ≥5 years. Antibiotic use in the seven days before admission was reported by 19.6% of participants overall, including 5.7% of those enrolled in Kakuma and 23.0–29.3% of those enrolled at the other three sites.

The household characteristics of AFI cases varied across sites (Table 1). For example, the main water source for the household was piped water in Kakuma (94.0%), Nairobi (77.1%) and Mombasa (72.2%), while in Kakamega it was a river or spring (43.4%). The main type of toilet/sanitary facility was pit latrine in Kakuma (91.7%) and Kakamega (90.4%) and was flush toilet in Nairobi (60.5%) and Mombasa (60.5%). The use of firewood as the main cooking fuel was relatively infrequent in Nairobi (4.5%) and Mombasa (4.7%) but was most common in Kakuma (80.8%) and Kakamega (50.0%). The most common animal the participants reported contact with in the prior two weeks was with poultry in Kakuma (7.9%), Kakamega (39.6%) and with cats in Mombasa (9.3%) and Nairobi (6.6%). The proportion of cases that reported sleeping under a mosquito net regularly ranged from 63.8% in Nairobi to 91.1% in Mombasa.

**Table 1. Demographic, clinical, and household characteristics of acute febrile illness (AFI) and undifferentiated fever (UF) cases, at four hospitals in Kenya, June 2017-March 2019[*].**

| | AFI cases by site | | | | UF and non-UF cases (all sites) | | Overall AFI N = 3,232 n (%) |
|---|---|---|---|---|---|---|---|
| | Kakuma n = 951 n (%) | Kakamega n = 396 n (%) | Nairobi n = 1140 n (%) | Mombasa n = 745 n (%) | UF cases n = 1735 n (%) | non-UF cases n = 1497 n (%) | |
| *Demographics* | | | | | | | |
| Male | 468 (49.2) | 223 (56.3) | 656 (57.5) | 426 (57.2) | 931 (53.7) | 842 (56.3) | 1774 (54.9) |
| Age group | | | | | | | |
| <12 months | 276 (29.0) | 91 (23.0) | 601 (52.7) | 297 (39.9) | 416 (24.0) | 849 (56.7) | 1265 (39.1) |
| 12–23 months | 176 (18.5) | 65 (16.4) | 241 (21.1) | 180 (24.2) | 282 (16.3) | 380 (25.4) | 662 (20.5) |
| 24–59 months | 161 (16.9) | 116 (29.3) | 183 (16) | 142 (19.1) | 441 (25.4) | 161 (10.8) | 602 (18.6) |
| 5–17 years | 119 (12.5) | 87 (22.0) | 74 (6.5) | 64 (8.6) | 299 (17.2) | 45 (3.0) | 344 (10.6) |
| 18–50 years | 202 (21.2) | 30 (7.6) | 37 (3.3) | 48 (6.4) | 264 (15.2) | 53 (3.5) | 317 (9.8) |
| 50+ years | 17 (1.8) | 7 (2) | 4 (0.4) | 14 (2.0) | 33 (1.9) | 9 (0.6) | 42 (1.3) |
| *Clinical characteristics* | | | | | | | |
| Symptoms | | | | | | | |
| Chills | 91 (9.6) | 70 (17.7) | 62 (5.4) | 18 (2.4) | 205 (30.2) | 36 (15.4) | 241 (7.5) |
| Lack of appetite | 95 (10.0) | 79 (19.9) | 168 (14.7) | 50 (6.7) | 281 (41.4) | 111 (0.5) | 392 (12.1) |
| Sore muscles | 42 (4.4) | 4 (1.0) | 31 (2.7) | 42 (5.6) | 109 (16.1) | 10 (4.3) | 119 (3.7) |
| Headache | 185 (19.5) | 68 (17.2) | 52 (4.6) | 67 (9.0) | 324 (47.7) | 48 (20.5) | 372 (11.5) |
| Cough | 546 (57.4) | 224 (56.6) | 655 (57.5) | 423 (56.8) | 825 (47.6) | 1023 (68.3) | 1848 (57.0) |
| Difficulty breathing | 156 (16.4) | 78 (19.7) | 513 (45.0) | 290 (38.9) | 279 (16.1) | 758 (50.6) | 1037 (32.1) |
| Shortness of breath | 35 (3.7) | 7 (1.8) | 122 (10.7) | 144 (19.3) | 81 (4.7) | 227 (15.2) | 308 (9.5) |
| Sore throat | 15 (1.6) | 3 (0.8) | 21 (1.8) | 2 (0.3) | 36 (5.3) | 5 (2.1) | 41 (1.3) |
| Runny nose | 108 (11.4) | 76 (19.2) | 261 (22.9) | 41 (5.5) | 239 (13.8) | 247 (16.5) | 486 (15.0) |
| Vomiting | 402 (42.3) | 160 (40.4) | 383 (33.6) | 280 (37.6) | 540 (31.1) | 685 (45.8) | 1225 (37.9) |
| Diarrhea | 393 (41.3) | 148 (37.4) | 439 (38.5) | 310 (41.6) | 198 (11.4) | 1092 (73.0) | 1290 (39.9) |
| Rash | 13 (1.4) | 12 (3.0) | 97 (8.5) | 75 (10.1) | 124 (7.2) | 73 (4.9) | 197 (6.1) |
| Physical exam findings | | | | | | | |
| Impaired consciousness | 11 (1.1) | 28 (7.1) | 112 (9.8) | 52 (7.0) | 88 (5.1) | 115 (7.7) | 203 (6.3) |
| Lethargy | 25 (2.6) | 292 (73.7) | 562 (49.3) | 358 (48.1) | 553 (31.9) | 684 (45.7) | 1237 (38.3) |
| Tachypnea | 272 (28.6) | 104 (26.3) | 361 (31.7) | 265 (35.6) | 453 (26.1) | 549 (36.7) | 1002 (31.0) |
| Rales or crackles | 144 (15.1) | 93 (23.5) | 372 (32.6) | 280 (37.6) | 295 (17.0) | 594 (39.7) | 889 (27.5) |
| Wheezing | 11 (1.1) | 6 (1.5) | 41 (3.6) | 21 (2.8) | 21 (1.2) | 58 (3.9) | 79 (2.4) |
| Oxygen saturation <90% on room air[†] | 26 (2.7) | 26 (6.6) | 195 (17.1) | 34 (4.6) | 96 (5.5) | 185 (12.4) | 281 (8.7) |
| Hepatomegaly | 8 (0.8) | 11 (2.8) | 36 (3.2) | 16 (2.1) | 39 (2.3) | 32 (2.1) | 71 (2.2) |
| Splenomegaly | 18 (1.9) | 28 (7.1) | 19 (1.7) | 14 (1.9) | 62 (3.6) | 17 (1.1) | 79 (2.4) |
| Rash | 12 (1.3) | 5 (1.3) | 25 (2.2) | 34 (4.6) | 52 (3.0) | 24 (1.6) | 76 (2.4) |
| Jaundice | 7 (0.7) | 14 (3.5) | 80 (7.0) | 32 (4.3) | 98 (5.7) | 35 (2.3) | 133 (4.1) |
| Discharge diagnosis[‡] | | | | | | | |
| Pneumonia | 359 (39.5) | 87 (24.5) | 380 (38.5) | 239 (36.0) | 366 (23.4) | 599 (44.7) | 1065 (36.6) |
| Malaria | 254 (28.0) | 203 (57.2) | 65 (6.6) | 96 (14.5) | 494 (31.5) | 130 (9.7) | 618 (21.2) |
| Gastroenteritis | 165 (18.2) | 44 (12.4) | 180 (18.3) | 185 (27.9) | 77 (4.9) | 498 (37.2) | 574 (19.7) |
| Meningitis | 4 (0.4) | 33 (9.3) | 122 (12.4) | 83 (13.0) | 153 (9.8) | 91 (6.8) | 242 (8.3) |
| Febrile convulsions | 10 (1.1) | 9 (2.5) | 129 (13.1) | 63 (9.5) | 155 (9.9) | 56 (4.2) | 211 (7.2) |
| Neonatal sepsis | 0 | 4 (1.1) | 139 (14.1) | 7 (1.0) | 95 (6.1) | 54 (4.0) | 150 (5.1) |
| Outcome (n = 3021) | | | | | | | |
| Death among all cases | 16 (1.7) | 12 (3.2) | 75 (7.3) | 28 (4.1) | 40 (2.6) | 74 (5.5) | 131 (4.3) |

*(Continued)*

**Table 1.** (Continued)

| | AFI cases by site | | | | UF and non-UF cases (all sites) | | Overall AFI N = 3,232 n (%) |
|---|---|---|---|---|---|---|---|
| | Kakuma n = 951 n (%) | Kakamega n = 396 n (%) | Nairobi n = 1140 n (%) | Mombasa n = 745 n (%) | UF cases n = 1735 n (%) | non-UF cases n = 1497 n (%) | |
| Death among aged <5 years | 9 (1.0) | 8 (2.1) | 69 (6.7) | 20 (2.9) | 26 (1.7) | 70 (5.2) | 106 (4.5) |
| Death among aged > = 5 years | 7 (0.7) | 4 (1.1) | 6 (0.6) | 8 (1.2) | 14 (0.9) | 4 (0.3) | 25 (3.8) |
| Past medical history | | | | | | | |
| Malnutrition | 62 (6.5) | 20 (5.1) | 148 (13.0) | 50 (6.7) | 110 (6.3) | 170 (11.4) | 280 (8.7) |
| Asthma | 17 (1.8) | 18 (4.6) | 21 (1.8) | 26 (3.5) | 52 (3.0) | 30 (2.0) | 82 (2.5) |
| TB under treatment | 11 (1.2) | 5 (1.3) | 24 (2.1) | 13 (1.7) | 30 (1.7) | 23 (1.5) | 53 (1.6) |
| TB (previously treated) | 15 (1.6) | 2 (0.5) | 24 (2.1) | 9 (1.2) | 33 (1.9) | 17 (1.1) | 50 (1.6) |
| Other chronic respiratory disease | 2 (0.2) | 3 (0.8) | 39 (3.4) | 53 (7.1) | 38 (2.2) | 59 (3.9) | 97 (3.0) |
| Immunodeficiency, including HIV | 10 (1.1) | 11 (2.8) | 43 (3.8) | 19 (2.6) | 53 (3.1) | 30 (2.0) | 83 (2.6) |
| Heart disease | 9 (1.0) | 3 (0.8) | 54 (4.7) | 20 (2.7) | 35 (2.0) | 51 (3.4) | 86 (2.7) |
| Diabetes | 10 (1.1) | 3 (0.8) | 10 (0.9) | 12 (1.6) | 32 (1.8) | 3 (0.2) | 35 (1.1) |
| Preterm (among aged <1 year) | 31/274 (11.3) | 15/90 (16.7) | 66/600 (11.0) | 27/294 (9.2) | 48/415 (11.6) | 91/848 (10.7) | 139/1258 (11.0) |
| Hospitalized in the past 12 months | 276 (29.0) | 76 (19.2) | 330 (29.0) | 108 (14.5) | 412 (23.8) | 378 (25.3) | 790 (24.4) |
| Hospitalized in past 12 months among <5 years | 199 (20.9) | 52 (13.1) | 291 (25.5) | 81 (10.9) | 271 (15.6) | 352 (23.5) | 623 (19.3) |
| Hospitalized in past 12 months among ≥5 years | 77 (8.1) | 24 (6.1) | 39 (3.4) | 27 (3.6) | 141 (8.1) | 26 (1.7) | 167 (5.2) |
| Reported antibiotic use in the 7 days before admission | 54 (5.7) | 116 (29.3) | 292 (25.6) | 171 (23.0) | 292 (16.8) | 341 (22.8) | 633 (19.6) |
| *Household characteristics* | | | | | | | |
| Mean number of people in household | 6 (SD = 3.0) | 5 (SD = 2.2) | 4 (SD = 1.3) | 4 (SD = 1.7) | 5 (SD = 2.5) | 5 (2.1) | 5 (SD = 2.3) |
| Mean number of people per sleeping room | 4.7 (SD = 2.3) | 2.8 (SD = 1.2) | 3.1 (SD = 1.3) | 3.6 (SD = 1.5) | 3.6 (SD = 1.9) | 3.7 (1.7) | 3.7 (SD = 1.8) |
| Main water source | | | | | | | |
| Piped water | 894 (94.0) | 148 (37.4) | 879 (77.1) | 538 (72.2) | 1305 (75.2) | 1154 (77.1) | 2459 (76.1) |
| Bought water | 3 (0.3) | 1 (0.3) | 128 (11.2) | 129 (17.3) | 119 (6.9) | 142 (9.5) | 261 (8.1) |
| River or spring | 26 (2.7) | 172 (43.4) | 22 (1.9) | 5 (0.7) | 153 (8.8) | 72 (4.8) | 225 (7.0) |
| Well | 3 (0.3) | 69 (17.4) | 36 (3.2) | 71 (9.5) | 99 (5.7) | 80 (5.3) | 179 (5.5) |
| Rain water/other | 25 (2.6) | 6 (1.5) | 75 (6.6) | 2 (0.3) | 57 (3.4) | 49 (3.3) | 108 (3.3) |
| Main type of toilet/sanitary facility | | | | | | | |
| Pit latrine | 871 (91.7) | 358 (90.4) | 432 (38.0) | 289 (38.8) | 1120 (64.6) | 830 (55.6) | 1950 (60.4) |
| Flush toilet | 1 (0.1) | 38 (9.6) | 703 (61.8) | 450 (60.5) | 557 (32.1) | 635 (42.5) | 1192 (36.9) |
| Bush/other | 78 (8.2) | 0 | 3 (0.3) | 5 (0.7) | 57 (0.3) | 29 (0.2) | 86 (2.7) |
| Main Type of fuel used for cooking | | | | | | | |
| Gas | 2 (0.2) | 54 (13.6) | 732 (64.2) | 278 (37.3) | 521 (30.0) | 545 (36.4) | 1066 (33.0) |
| Firewood | 768 (80.8) | 198 (50.0) | 51 (4.5) | 35 (4.7) | 653 (37.6) | 399 (26.7) | 1052 (32.5) |
| Charcoal | 176 (18.5) | 131 (33.1) | 106 (9.3) | 270 (36.2) | 372 (21.4) | 311 (20.8) | 683 (21.1) |
| Kerosine | 2 (0.2) | 10 (2.5) | 233 (20.4) | 160 (21.5) | 175 (10.1) | 230 (15.4) | 405 (12.5) |
| Electricity/other | 3 (0.3) | 3 (0.8) | 18 (1.6) | 2 (0.3) | 14 (0.8) | 12 (0.8) | 26 (0.8) |
| Electricity in home | 116 (12.2) | 228 (57.6) | 1065 (93.4) | 683 (91.7) | 1043 (60.1) | 1049 (70.1) | 2092 (64.7) |
| Exposures during 2 weeks before hospitalization | | | | | | | |
| Person with fever | 41 (4.3) | 62 (15.7) | 141 (12.4) | 52 (7.0) | 155 (8.9) | 141 (9.4) | 296 (9.2) |
| Travel outside district of residence | 14 (1.5) | 27 (6.8) | 182 (16.0) | 88 (11.8) | 172 (9.9) | 139 (9.3) | 311 (9.6) |
| Contact with animals | | | | | | | |
| Poultry | 75 (7.9) | 157 (39.6) | 46 (4.0) | 40 (5.4) | 240 (13.8) | 78 (5.2) | 318 (9.8) |
| Cat | 41 (4.3) | 126 (31.8) | 75 (6.6) | 69 (9.3) | 222 (12.8) | 89 (5.9) | 311 (9.6) |
| Dog | 11 (1.2) | 65 (16.4) | 33 (2.9) | 19 (2.6) | 96 (5.5) | 32 (2.1) | 128 (4.0) |
| Cow | 3 (0.3) | 93 (23.5) | 13 (1.1) | 7 (0.9) | 92 (5.3) | 24 (1.6) | 116 (3.6) |

(*Continued*)

**Table 1.** (Continued)

| | AFI cases by site | | | | UF and non-UF cases (all sites) | | Overall AFI N = 3,232 n (%) |
|---|---|---|---|---|---|---|---|
| | **Kakuma n = 951 n (%)** | **Kakamega n = 396 n (%)** | **Nairobi n = 1140 n (%)** | **Mombasa n = 745 n (%)** | **UF cases n = 1735 n (%)** | **non-UF cases n = 1497 n (%)** | |
| Goat | 20 (2.1) | 22 (5.6) | 15 (1.3) | 11 (1.5) | 55 (3.2) | 13 (0.9) | 68 (2.1) |
| Pigs, sheep and other | 10 (1.1) | 21 (5.3) | 13 (1.1) | 6 (0.8) | 45 (2.6) | 5 (0.3) | 50 (1.5) |
| Sleep under mosquito net | 783 (82.3) | 360 (90.9) | 727 (63.8) | 679 (91.1) | 1375 (79.3) | 1174 (78.4) | 2549 (78.9) |

<sup>*</sup> Missing values excluded from denominators

† Excluded values not measured on room air: 1 (<1%) in Kakuma, 15 (4%) in Kakamega, 92 (8%) in Nairobi and 35 (5%) in Mombasa

‡ Present diagnoses recorded in ≥5% of AFI cases (complete list of discharge diagnoses presented in S1 Table); diagnoses not mutually exclusive, cases could have more than one discharge diagnosis

Among 1,314 UF cases with blood samples tested by TAC, 715 (54.4%) had no pathogens detected (Fig 2 and S2 Table). Cases aged <12 months had the highest proportion of cases with no pathogen detected (75.0%, 147/196), and those aged 18–49 years had the lowest proportion (39.0%, 92/236) (Fig 3A and S3 Table). The Nairobi site had the highest proportion of cases with no pathogen detected (79.7%, 248/311), and the Kakamega site had the lowest proportion (42.3%, 94/222) (Fig 3B and S4 Table).

The most common pathogen identified among UF cases was *Plasmodium* spp., detected as a single pathogen in 471 (35.8%) samples and in combination with other pathogens in 38 (2.9%) of samples. *Plasmodium* was the most frequently detected pathogen across all ages (Fig 3A and S3 Table) and all sites (Fig 3B and S4 Table). The frequency of *Plasmodium* detection among UF cases by site was 52.7% (117/222) in Kakamega, 45.2% (233/515) in Kakuma, 42.1% (112/266) in Mombasa, and 15.1% (47/311) in Nairobi. The age groups with the highest proportion of *Plasmodium* detected were 5–17 years (51.3%, 139/271) and 18–49 years (50.4%, 119/236); the lowest portion was among age <1 year (21.4%, 42/196).

Chikungunya virus was found in 2.3% (30/1314) of UF cases (22 chikungunya alone and 8 together with *Plasmodium*). Chikungunya was detected only in the Mombasa site among cases aged <50 years. Cases with chikungunya were first detected in December 2017 (the same month that surveillance started in Mombasa). The highest number of monthly cases with chikungunya detected was in January 2018 (n = 13); all other months had <5 cases detected (Fig 4). Other pathogens detected by TAC were present in <1% of UF cases.

Among 776 UF cases with blood culture performed, 11 (1.4%) grew potentially pathogenic isolates, including *Staphylococcus aureus* (n = 5), *Streptococcus pneumoniae* (n = 1), *Escherichia coli* (n = 1), *Salmonella* group B (n = 1), *Salmonella enterica* serovar Typhi (n = 1), and *Cryptococcus* species (n = 1) (Table 2).

## Discussion

Across four ecologically distinct sites in Kenya the majority of cases enrolled in AFI (>75%) and UF (>65%) surveillance were children aged <5 years, among whom there was a case fatality of 4.5%. Case fatality varied by site, and was highest in Nairobi (7.3%), likely reflecting the severity of cases admitted to the national referral hospital. *Plasmodium* was the most common pathogen detected among UF cases across all sites and age groups. More than half of UF had no pathogens detected by TAC, highlighting the challenges of characterizing the causes of undifferentiated fever.

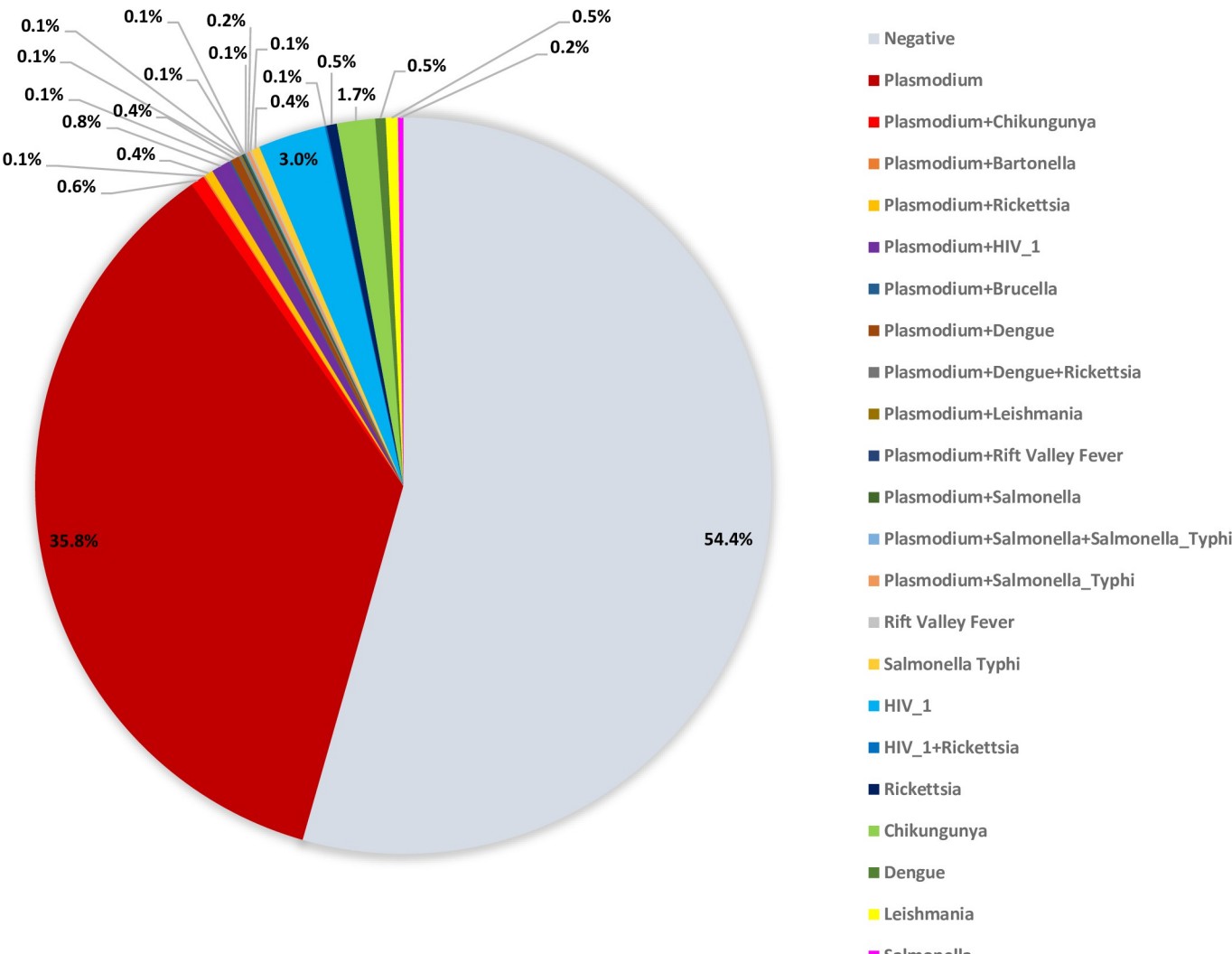

**Fig 2. Pathogens detected by TAC among UF cases (n = 1314) in Kenya at four hospitals in Kenya, June 2017-March 2019.**

While there were important reductions in malaria morbidity and mortality globally, and particularly in Sub-Saharan Africa, from 2000–2015, key indicators have not improved since 2015, and malaria remains a major cause of morbidity and mortality. In our study, malaria was the most frequently detected pathogen among persons in AFI surveillance across all age groups and sites. The age groups with the highest frequency of *Plasmodium* detection were 5–17 and 18–49 years; the finding that school-aged children and adolescents harbored the highest prevalence of infection is consistent with population-based prevalence studies from the highly endemic Lake Region of western Kenya, such as Kakamega [19]. In these settings, and to a lesser extent the Mombasa site on the Kenya coast, repeated exposure to malaria infection results in naturally acquired or partial immunity, which manifests in a dampening of symptoms during each subsequent infection [20]. Among older children and adults, naturally acquired immunity results in the ability to harbor low-density asymptomatic infections, many of which may only be detected through molecular diagnostics [21]. This results in a higher burden of malaria infections requiring hospitalization among young children. In areas of low endemicity, or which are subject only to epidemics, such as the Kakuma site [15], or infections

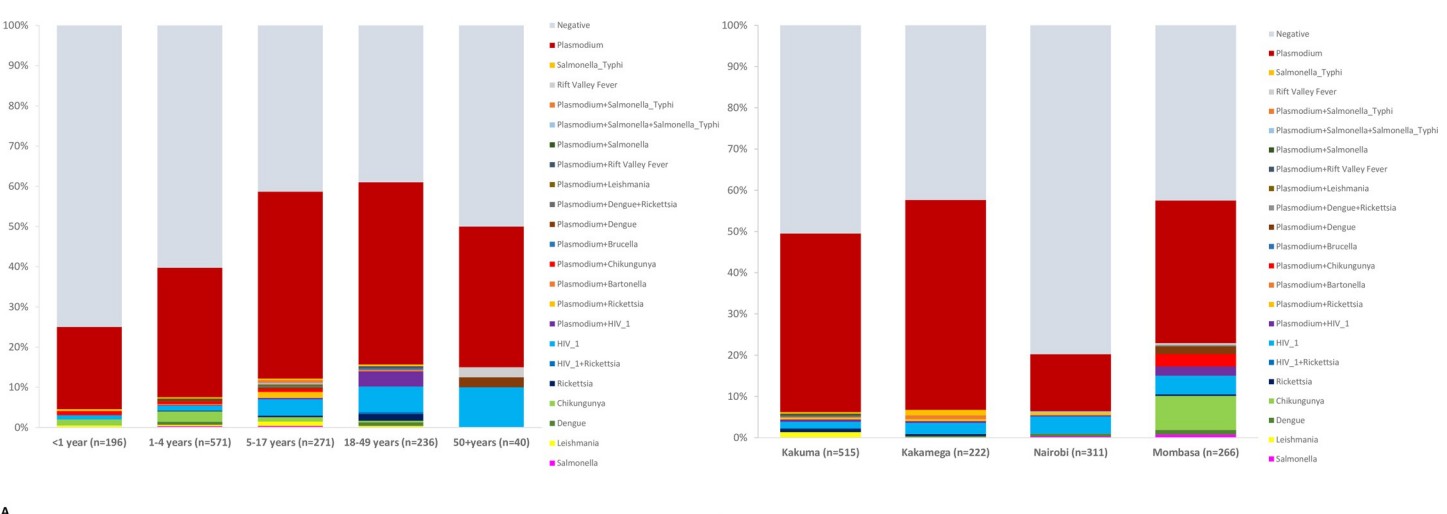

**Fig 3.** Pathogens detected by TAC among UF cases (n = 1,314) by (A) age group and (B) site, June 2017-March 2019. HIV was detected in 3.9% (51/1314) of UF cases tested by TAC (39 HIV alone, 11 together with *Plasmodium*, and 1 together with *Rickettsia*). HIV was more common among adult UF cases (25/236 [10.6%] in ages 18–49, 4/40 [10.0%] in ages ≥50 years) than among children (2/196 [0.5%] in age <1 year, 8/571 [1.4%] in ages 1–4 years, and 12/271 [4.4%] in ages 5–17 years). Across sites, detection of HIV ranged from 2.3% (12/515) in Kakuma to 6.8% (18/266) in Mombasa.

acquired during travel, such as the Nairobi site, older children and adults have not developed naturally acquired immunity, infections result in the development of high parasite densities and more severe symptoms, and the burden of hospitalization is more proportionally

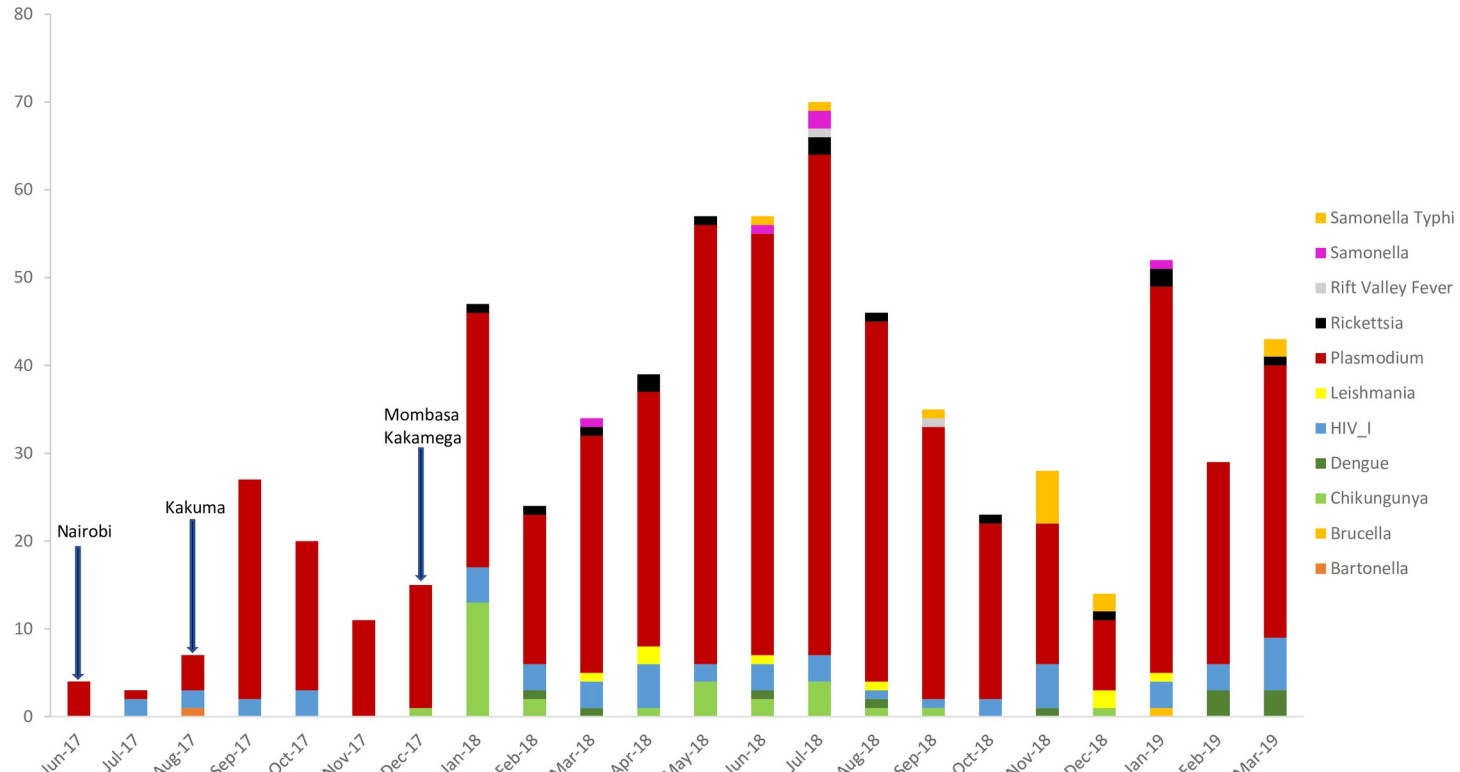

**Fig 4. Pathogens detected by TAC among UF cases (n = 1,314) by month, across four hospitals in Kenya, June 2017- March 2019.** Arrows indicate surveillance start date. Counts reflect positive test results; some cases have more than one positive result.

**Table 2. Blood culture results from undifferentiated fever cases at four hospitals in Kenya, June 2017-March 2019.**

| Pathogens isolated from blood culture | Kakuma n (%) n = 339 | Kakamega n (%) n = 156 | Nairobi n (%) n = 258 | Overall n (%) n = 753 |
|---|---|---|---|---|
| *Staphylococcus aureus* | 1 (0.3) | 2 (1.3) | 3 (1.2) | 6 (0.8) |
| *Streptococcus pneumoniae* | 0 | 1 (0.6) | 0 | 1 (0.1) |
| *Escherichia Coli* | 0 | 0 | 1 (0.4) | 1 (0.1) |
| *Salmonella* group B | 1 (0.3) | 0 | 0 | 1 (0.1) |
| *Salmonella enterica* serovar Typhi | 0 | 1 (0.6) | 0 | 1 (0.1) |
| *Cryptococcus* spp. | 0 | 0 | 1 (0.4) | 1 (0.1) |

distributed across age groups. In the context of a large proportion of UF participants in whom we did not identify an underlying pathogen, our findings of higher prevalence of malaria among admitted UF patients in Kakamega and Mombasa may be an incidental finding and a reflection of our use of a molecular test in a setting of high rates of low-density asymptomatic parasite infections. Ascertainment of parasite densities may have allowed us to estimate the proportion of these infections in which malaria was in the causality pathway for hospitalization. However, in Nairobi and in Kakuma, findings of high rates of malaria-associated hospitalization are consistent with prior studies [21]. Our findings of high rates of *Plasmodium* positivity are also consistent with other studies among children seeking care for fever in sub-Saharan Africa where *P. falciparum* was estimated to be the cause of fever in 37.7%, with substantial variability over time and place [22]. While many AFI studies exclude patients with malaria to focus on non-malarial etiologies of fever, our study highlights the importance of capturing data on malaria, including co-detection of malaria together with other pathogens, as well as the complexities of interpreting *Plasmodium* positivity across diverse etiologic contexts.

After *Plasmodium*, the next most common pathogen detected among UF cases was HIV. Although rarely detected among children, approximately one of every ten adult UF cases was HIV-infected. As with *Plasmodium*, detection of HIV among UF cases does not necessarily imply that it is the cause of the fever. Fever is a common presenting sign of acute HIV infection [23], however, immunosuppression resulting from chronic HIV infection can also increase the risk for other infectious diseases that may also cause fever [24]. Based on the TAC assay, it is not possible to distinguish between acute and chronic HIV; further testing is needed to better characterize the role of HIV in AFI and UF in adults. There is evidence of declining HIV incidence in Sub-Sahara Africa, including in the East African region and Kenya [25, 26]. Nonetheless, HIV should be considered in adult patients presenting with fever in Kenya, either as an acute cause of febrile illness or as an underlying cause.

Reports of chikungunya outbreaks from African countries have increased over the past 20 years [27]. An outbreak of chikungunya virus occurred in Mombasa from late 2017 through mid-2018, as reflected in these surveillance data; genomic analyses of the implicated strain detected mutations that could make transmission more efficient, including potential adaptation to *Aedes albopictus*, while prior chikungunya outbreaks in Kenya have been spread only by *Ae. aegypti* [28]. A population-based cohort study conducted in 2014–2018 in Kilifi, coastal Kenya found that 12.7% of febrile children aged <16 years tested positive for chikungunya, with evidence of endemic disease occurring during non-epidemic periods [11]. Another study of AFI in children aged <18 years in two sites in Kenya in 2014–2015 reported detection of chikungunya in 5.5% of cases in sites in the coastal region and 13.1% of cases in two sites near Lake Victoria in Western Kenya [29]. Our surveillance did not detect chikungunya cases outside of the coastal region (Mombasa site), or outside the time period of a recognized outbreak

(late December 2017 to mid-2018), and the percentage of cases with chikungunya detected was lower than reported in these other studies in Kenya [11, 29]. Nonetheless, these data in conjunction highlight the importance of chikungunya as a cause of fever as well as the need for continued surveillance to inform control strategies.

Our data highlight some of the challenges in determining the etiology of AFI. With so many viral, bacterial, and fungal pathogens potentially causing fever, frequently no etiology is determined for AFI cases, even after extensive diagnostic testing [5, 7, 30]. AFI TAC tests for 28 distinct pathogens; however, many of the pathogens are found more commonly in adults while the AFI and UF cases enrolled were predominantly children, as reflected in the proportion with no pathogen detected across age groups. A study of febrile children aged <5 years presenting for outpatient care in Tanzania using metagenomic next generation sequencing of blood samples found that half of the children had evidence of at least one virus, most commonly enteroviruses, rotaviruses and human herpesvirus 6 [31]; none of these pathogens were included on the TAC card used for our surveillance, and the UF case definition used to target TAC testing excluded diarrheal AFI cases. Furthermore, although we did not focus on the detection of respiratory pathogens and excluded those with evidence of lower respiratory tract infection from TAC testing, respiratory symptoms and pneumonia diagnosis were common among UF cases, suggesting that collection of a respiratory sample and testing for respiratory pathogens may have yielded more etiologic information. We detected very few pathogens detected by blood culture. Blood culture is the gold standard for invasive bacterial disease detection but is insensitive, particularly when antibiotics have been previously administered [5].

This study was subject to several additional limitations. Enrollment and sample collection was suboptimal, potentially leading to bias by not including certain groups of patients. Some of the sites were referral hospitals, and patients may have received treatment before arrival which could affect the sensitivity of diagnostic methods used. Allowing for symptom onset up to 13 days before presentation to the hospital could have limited the detection of pathogens most easily detected early in the course of illness. We had limited ability to examine seasonality or cyclical patterns of AFI pathogens, due to a relatively short study period and detection of small numbers of cases for most pathogens. We could not examine associations between pathogens and outcomes due to small case counts for most pathogens detected. Finally, as noted above for *Plasmodium* and HIV, detection of the pathogen does not necessarily reflect the AFI cause.

Surveillance at four ecologically distinct sites across Kenya showed that most patients hospitalized with fever are young children, and the most common pathogen detected across all sites and age groups was *Plasmodium*, highlighting the importance of malaria among febrile illnesses. More than half of cases with blood samples collected had no pathogen detected, despite testing for 28 causes of fever, reflecting the challenge of determining the etiology of fever. Malaria prevention and control efforts are critical for reducing the burden of AFI, and improved diagnostic testing is needed to provide better insight into non-malarial causes of fever. The high case fatality of AFI underscores the need to optimize diagnosis and appropriate management of AFI.

## Supporting information

**S1 Fig. Map of surveillance sites.** Sites included Kenyatta National Hospital in Nairobi City County, Coast General Teaching and Referral Hospital in Mombasa County, Kakamega County Referral Hospital in Kakamega County, and Kakuma Refugee Camp General Hospital in Turkana County. Size of circle reflects bed capacity of the participating hospital in each site. (DOCX)

**S2 Fig. TAC card targets.**
(DOCX)

**S1 Table. Discharge diagnoses among acute febrile illness (AFI) cases, at four hospitals in Kenya, June 2017-March 2019.**
(DOCX)

**S2 Table. Pathogens detected by TAC among UF cases (n = 1314) in Kenya at four hospitals in Kenya, June 2017-March 2019.**
(DOCX)

**S3 Table. Pathogens detected by TAC among UF cases (n = 1,314) by age group, June 2017-March 2019.**
(DOCX)

**S4 Table. Pathogens detected by TAC among UF cases (n = 1,314) by site, June 2017-March 2019.**
(DOCX)

## Acknowledgments

We thank the teams of surveillance officers and laboratory technicians who generated the data presented in this paper. We also thank the participants for agreeing to share information and samples even while acutely ill, and the clinical teams in participating hospitals for their cooperation and support.

**Disclaimer:** The findings and conclusions in this paper are those of the authors and do not necessarily represent the views of the U.S. Centers for Disease Control and Prevention.

## Author Contributions

**Conceptualization:** Jennifer R. Verani, Elizabeth A. Hunsperger, Godfrey Bigogo, Aaron M. Samuels, Kariuki Njenga, Joel M. Montgomery, Marc-Alain Widdowson.

**Data curation:** Eric Ng' eno, Doris Marwanga, Derrick Amon, Melvin Ochieng.

**Formal analysis:** Jennifer R. Verani, Doris Marwanga.

**Funding acquisition:** Joel M. Montgomery, Marc-Alain Widdowson.

**Investigation:** Jennifer R. Verani, Eric Ng' eno, Elizabeth A. Hunsperger, Peninah Munyua, Eric Osoro, Godfrey Bigogo, Derrick Amon, Melvin Ochieng.

**Methodology:** Jennifer R. Verani, Eric Ng' eno, Elizabeth A. Hunsperger, Peninah Munyua, Eric Osoro, Godfrey Bigogo, Aaron M. Samuels, Kariuki Njenga, Joel M. Montgomery, Marc-Alain Widdowson.

**Project administration:** Jennifer R. Verani, Eric Ng' eno, Eric Osoro, Joel M. Montgomery.

**Resources:** Paul Etau, Victor Bandika, Victor Zimbulu, John Kiogora, John Wagacha Burton, Joel M. Montgomery, Marc-Alain Widdowson.

**Supervision:** Jennifer R. Verani, Eric Ng' eno, Elizabeth A. Hunsperger, Peninah Munyua, Eric Osoro, Paul Etau, Victor Bandika, Victor Zimbulu, John Kiogora, Kariuki Njenga, Joel M. Montgomery, Marc-Alain Widdowson.

**Visualization:** Jennifer R. Verani, Doris Marwanga.

**Writing – original draft:** Jennifer R. Verani, Elizabeth A. Hunsperger, Peninah Munyua.

**Writing – review & editing:** Jennifer R. Verani, Eric Ng' eno, Elizabeth A. Hunsperger, Peninah Munyua, Eric Osoro, Doris Marwanga, Godfrey Bigogo, Derrick Amon, Melvin Ochieng, Paul Etau, Victor Bandika, Victor Zimbulu, John Kiogora, John Wagacha Burton, Emmanuel Okunga, Aaron M. Samuels, Kariuki Njenga, Joel M. Montgomery, Marc-Alain Widdowson.

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
