## [Decision Letter · Decision Letter 0]

25 Mar 2024

PONE-D-24-01611Acute febrile illness in Kenya: clinical characteristics and pathogens detected among patients hospitalized with fever, 2017-2019PLOS ONE

Dear Dr. Verani,

Thank you for submitting your manuscript to PLOS ONE. After careful consideration, we feel that it has merit but does not fully meet PLOS ONE’s publication criteria as it currently stands. Therefore, we invite you to submit a revised version of the manuscript that addresses the points raised during the review process. Please answer within the rebutal letter and IN the text of the revised article please include all the requested informations.

We look forward to receiving your revised manuscript.

Kind regards,

Pierre Roques, Ph.D.

Academic Editor

PLOS ONE

Journal Requirements:

4. We note you have included a table to which you do not refer in the text of your manuscript. Please ensure that you refer to Table 2 in your text; if accepted, production will need this reference to link the reader to the Table.

5. We note that [Figure S1] in your submission contain [map/satellite] images which may be copyrighted. All PLOS content is published under the Creative Commons Attribution License (CC BY 4.0), which means that the manuscript, images, and Supporting Information files will be freely available online, and any third party is permitted to access, download, copy, distribute, and use these materials in any way, even commercially, with proper attribution. For these reasons, we cannot publish previously copyrighted maps or satellite images created using proprietary data, such as Google software (Google Maps, Street View, and Earth). For more information, see our copyright guidelines: http://journals.plos.org/plosone/s/licenses-and-copyright.

a. You may seek permission from the original copyright holder of Figure S1 to publish the content specifically under the CC BY 4.0 license.  

Reviewers' comments:

Reviewer's Responses to Questions

**Comments to the Author**

1. Is the manuscript technically sound, and do the data support the conclusions?

Reviewer #1: Yes

2. Has the statistical analysis been performed appropriately and rigorously? 

Reviewer #1: Yes

3. Have the authors made all data underlying the findings in their manuscript fully available?

Reviewer #1: No

4. Is the manuscript presented in an intelligible fashion and written in standard English?

Reviewer #1: Yes

5. Review Comments to the Author

Reviewer #1: General Comments

This is an expansive and interesting study that delves into the pathogens associated with acute febrile illness, and particularly “undifferentiated fever” (i.e. excluding diarrheal disease and/or lower respiratory disease) across four different geographical settings in Kenya. It is limited by the number of pathogens tested for and the fact that only blood specimens were collected and tested. Further depth could also have been undertaken with respect to analysing risk factors, i.e. for more common infections (malaria) as well as perhaps by grouping similar pathogens. However, the limitations are in fact an important component, and do a good job of highlighting the challenges of conducting comprehensive AFI surveillance – there are just too many pathogens that are associated with fever, and too many biological specimens that would need to be collected and tested to look for them all! There are enormous opportunities in standardizing methodologies and approaches for AFI surveillance and characterization, and this paper adds nicely to that literature.

Specific Comments

- Lines 74-78: When a patient is “admitted”, does this refer to both in-patient and out-patient care? The implication from the language is in-patient only, in which case it would be helpful to describe why individuals presenting with fever who were deemed eligible for out-patient care were not considered eligible. Or is this because eligibility was determined by review of logs, and so most out-patients would have already left the hospital by the time the surveillance staff tried to locate them? This could be worth clarifying, as it would help to explain the quite high percentage of patients that the surveillance staff were not able to locate. It could also have implications for some of the findings presented later, particularly with respect to mortality.

- Line 131: Typically, “assent” refers to verbal agreement; do the authors mean to say that verbal assent was documented in writing? Or should this be changed to “written consent” being obtained?

- Line 140: Was the lack of blood collection in 30% of cases due to refusal by the patient or for other reasons?

- Line 140-150: Were there any cases that were excluded due to signs of obvious injury as the cause of the infection (often an exclusion factor in AFI studies – see for example https://www.ncbi.nlm.nih.gov/pmc/articles/PMC9436081/), or other routine childhood infections not typically associated with transmissible pathogens (i.e. otitis)? The authors do note that a number of common and endemic viral causes of fever (i.e. enteroviruses, adenoviruses) were not considered here which is definitely a limitation, but there might be non-infectious causes that should be considered as well.

- Lines 161-170: I appreciate that most individual diseases were detected too infrequently for robust risk factor analysis, but presumably this could have been done for malaria? Likewise the authors could have considered grouping similar pathogens (i.e. arboviruses) to look for risk factors associated with pathogens with similar ecological drivers and transmission patterns.

- Line 195-196: Suggest highlighting Figure 4 and seasonal patterns more clearly in the results section, it’s a bit hidden here but is potentially important. I appreciate that the authors consider the data to be too sparse for specific pathogens to conduct seasonal analyses (see Line 281) but there are additional qualitative observations that could be made; also there are sufficient data on overall number of eligible patients over time that could potentially be used to look for seasonal trends, segregated by region, in number of AFI cases presenting to the hospitals.

- Line 227-229: This is an important point and one that perhaps could be highlighted further, particularly since quite a number of other studies focused on undifferentiated fever in malaria-endemic settings will exclude patients with a positive malaria RDT – despite the fact that sub-clinical malaria may abound and thus Plasmodium infection actually isn’t contributing to symptoms. Researchers working on AFI should be encouraged to include malaria-positive patients in their studies to ascertain if there might be co-infections causing AFI.

- Line 274: Blood culture can also potentially can be biased by prophylactic antibiotic usage; was treatment history or reported self-medication included on the intake questionnaire?

- Table 1: How was the questionnaire developed and validated? For animal exposure, contact with rodents (i.e. presence of rodents in the house; recent sightings of rodents) would seem to be an important variable to include; likewise perhaps having a separate category for other wildlife. How was “contact” with animals defined?

- Figure 2: This is a helpful chart to highlight the large proportion of cases in which no pathogen was detected, but not very helpful for visualizing the breakdown of identified pathogens/co-infections. Consider re-doing the chart to show only cases in which one or more pathogens were detected, to spread out the colors more evenly and allow readers better granularity of information. Figure 3 can remain as is to highlight the large proportion of cases with no pathogen detected (which is more interesting to see split across age groups, in any case).

- Supplemental tables: The supplemental materials do not seem to be available for review which is a shame, as I wanted to see what additional information was provided, especially Tables S1 and S2. In addition, the authors note assistance from the journal would be required to make underlying data available – one option would be to submit the manuscript, including underlying de-identified databases, to a pre-print server (i.e. https://www.medrxiv.org/) or online repository (i.e. https://osf.io/; also suitable for pre-prints).

6. PLOS authors have the option to publish the peer review history of their article (what does this mean?). If published, this will include your full peer review and any attached files.

Reviewer #1: **Yes: **Claire Standley

---

## [Author Response · Author response to Decision Letter 0]

7 May 2024

May 7, 2024

Dear Dr. Roques,

Many thanks for sharing these reviewer comments. We have revised the manuscript to address their concerns and have prepared a point-by-point response to each comment below. We have also addressed the additional journal requirements noted below. Please let us know if anything else is needed. Thank you again for your consideration of this manuscript. 

Regards,

Jennifer Verani

Author response: We have revised the file names and the title page to adhere to PLOS ONE’s style requirements – apologies for having missed that with the initial submission. 

Author response: Apologies for that inconsistency. We have aligned them in the resubmission. 

Author response: At the time of submission, we anticipated being able to make the data publicly available without restrictions. However, the Data Protection Act (TheDataProtectionAct__No24of2019.pdf (kenyalaw.org)), which was enacted in Kenya in 2019 and more strictly enforced in recent years, states that:

53. Research, history and statistics (1) The further processing of personal data shall be compatible with the purpose of collection if the data is used for historical, statistical or research purposes and the data controller or data processor shall ensure that the further processing is carried out solely for such purposes and will not be published in an identifiable form. (2) The data controller or data processor shall take measures to establish appropriate safeguards against the records being used for any other purposes.

The data presented in this manuscript have several stewards. The Kenya Medical Research Council (KEMRI) provided the primary ethical approval for the work (via the KEMRI Scientific and Ethical Review Unit) and owns the data. Washington State University was the implementing partner at the time of data collection and was responsible for safekeeping and management of the data. The Kenya Ministry of Health has been involved in the surveillance from its inception and is currently assuming leadership of the platform; therefore, moving forward the data will be co-owned by the Ministry of Health and KEMRI. After discussion among co-authors representing the various institutions and considering the Data Protection Act, we unfortunately cannot make the data publicly without any assurance that the data will be used in accordance with the aims of the protocol under which they were collected. The data can be made available through a process of managed access requiring the submission of a request for consideration by the KEMRI Data Governance Committee via email cghr@kemri.go.ke or telephone +254 (20) 22923.

We will revise our statement on data availability and request that you please seek input form an editor on a possible exception to this policy. 

4. We note you have included a table to which you do not refer in the text of your manuscript. Please ensure that you refer to Table 2 in your text; if accepted, production will need this reference to link the reader to the Table.

Author response: We have added a call-out to Table 2 in the relevant paragraph of the Results section. 

5. We note that [Figure S1] in your submission contain [map/satellite] images which may be copyrighted. All PLOS content is published under the Creative Commons Attribution License (CC BY 4.0), which means that the manuscript, images, and Supporting Information files will be freely available online, and any third party is permitted to access, download, copy, distribute, and use these materials in any way, even commercially, with proper attribution. For these reasons, we cannot publish previously copyrighted maps or satellite images created using proprietary data, such as Google software (Google Maps, Street View, and Earth). For more information, see our copyright guidelines: http://journals.plos.org/plosone/s/licenses-and-copyright.

Author response: Thank you for noting this issue with the copyrighted image. We have generated another map using R software that should not have copyright issues. 

a. You may seek permission from the original copyright holder of Figure S1 to publish the content specifically under the CC BY 4.0 license. 

Author response: We have added captions for the Supporting Information after the references in the manuscript, and have updated the in-text citations. 

Reviewers' comments:

Reviewer's Responses to Questions

Comments to the Author

1. Is the manuscript technically sound, and do the data support the conclusions?

Reviewer #1: Yes

2. Has the statistical analysis been performed appropriately and rigorously? 

Reviewer #1: Yes

3. Have the authors made all data underlying the findings in their manuscript fully available?

Reviewer #1: No

4. Is the manuscript presented in an intelligible fashion and written in standard English?

Reviewer #1: Yes

5. Review Comments to the Author

Reviewer #1: General Comments

This is an expansive and interesting study that delves into the pathogens associated with acute febrile illness, and particularly “undifferentiated fever” (i.e. excluding diarrheal disease and/or lower respiratory disease) across four different geographical settings in Kenya. It is limited by the number of pathogens tested for and the fact that only blood specimens were collected and tested. Further depth could also have been undertaken with respect to analysing risk factors, i.e. for more common infections (malaria) as well as perhaps by grouping similar pathogens. However, the limitations are in fact an important component, and do a good job of highlighting the challenges of conducting comprehensive AFI surveillance – there are just too many pathogens that are associated with fever, and too many biological specimens that would need to be collected and tested to look for them all! There are enormous opportunities in standardizing methodologies and approaches for AFI surveillance and characterization, and this paper adds nicely to that literature.

Author response: We thank the reviewer for their comments. 

Specific Comments

- Lines 74-78: When a patient is “admitted”, does this refer to both in-patient and out-patient care? The implication from the language is in-patient only, in which case it would be helpful to describe why individuals presenting with fever who were deemed eligible for out-patient care were not considered eligible. Or is this because eligibility was determined by review of logs, and so most out-patients would have already left the hospital by the time the surveillance staff tried to locate them? This could be worth clarifying, as it would help to explain the quite high percentage of patients that the surveillance staff were not able to locate. It could also have implications for some of the findings presented later, particularly with respect to mortality.

Author response: This study was restricted to hospitalized in-patients only. Although we initially considered enrolling both in-patients and out-patients, we did not have adequate resources to manage the logistics of comprehensive screening of all potentially eligible in-patients and out-patients. We chose to focus our efforts on the more severe manifestations of AFI, i.e. among persons requiring admission. The eligible hospitalized patients that could not be located were primarily patients that were discharged or transferred to another facility before the field team was able to enroll them. 

- Line 131: Typically, “assent” refers to verbal agreement; do the authors mean to say that verbal assent was documented in writing? Or should this be changed to “written consent” being obtained?

Author response: Assent refers to a willingness to participate in research given by persons too young to legally give informed consent but old enough to understand the proposed research, study procedures and potential risks and benefits. We obtained written assent (in addition to written consent from parent or guardian) for participants aged 7 to 17 years. 

- Line 140: Was the lack of blood collection in 30% of cases due to refusal by the patient or for other reasons?

Author response: Failure to obtain blood for culture was primarily due to either refusal or inability to obtain specimen; the latter was particularly a challenge in young children. 

- Line 140-150: Were there any cases that were excluded due to signs of obvious injury as the cause of the infection (often an exclusion factor in AFI studies – see for example https://www.ncbi.nlm.nih.gov/pmc/articles/PMC9436081/), or other routine childhood infections not typically associated with transmissible pathogens (i.e. otitis)? The authors do note that a number of common and endemic viral causes of fever (i.e. enteroviruses, adenoviruses) were not considered here which is definitely a limitation, but there might be non-infectious causes that should be considered as well.

Author response: We excluded patients for whom the primary reason for seeking care was injury or trauma. We have edited the methods section to reflect this exclusion criterion. We did not exclude children with acute otitis media (which is generally caused by transmissible pathogens, e.g. Streptococcus pneumoniae, Haemophilus influenzae and Moraxella catarrhalis). During the pilot phase of the surveillance, we did originally have an additional criterion for undifferentiated fever (UF) cases (i.e. those who had blood samples collected) that was “absence of a clear focus of fever based on history and physical examination” (which might have included acute otitis media) so that our etiologic testing would focus on patients without a clear cause for their fever; however we found this criterion very difficult to operationalize consistently and therefore collected blood from eligible UF cases even if infections such as acute otitis media were manifest. While it is possible that there might have been some non-infectious causes of fever (e.g. oncologic or rheumatologic diseases) among patients meeting the study AFI criteria, the vast majority of those enrolled had a discharge diagnosis for one or more infectious illnesses. 

- Lines 161-170: I appreciate that most individual diseases were detected too infrequently for robust risk factor analysis, but presumably this co

---

## [Decision Letter · Decision Letter 1]

5 Jun 2024

Acute febrile illness in Kenya: clinical characteristics and pathogens detected among patients hospitalized with fever, 2017-2019

PONE-D-24-01611R1

Dear Dr. Verani,

We’re pleased to inform you that your manuscript has been judged scientifically suitable for publication and will be formally accepted for publication once it meets all outstanding technical requirements.

Kind regards,

Pierre Roques, Ph.D.

Academic Editor

PLOS ONE

Additional Editor Comments (optional): I joint the reviewer about the question of survery of animals around the houses but also agreed that this is not mandatory in this article.

Reviewers' comments:

Reviewer's Responses to Questions

**Comments to the Author**

1. If the authors have adequately addressed your comments raised in a previous round of review and you feel that this manuscript is now acceptable for publication, you may indicate that here to bypass the “Comments to the Author” section, enter your conflict of interest statement in the “Confidential to Editor” section, and submit your "Accept" recommendation.

Reviewer #1: All comments have been addressed

2. Is the manuscript technically sound, and do the data support the conclusions?

Reviewer #1: Yes

3. Has the statistical analysis been performed appropriately and rigorously? 

Reviewer #1: Yes

4. Have the authors made all data underlying the findings in their manuscript fully available?

Reviewer #1: No

5. Is the manuscript presented in an intelligible fashion and written in standard English?

Reviewer #1: Yes

6. Review Comments to the Author

Reviewer #1: Many thanks for the detailed and considered responses to my comments - the manuscript will be an asset to the literature on AFI studies. While I support publication in its current form, if other reviewers or the editors request further edits, you could consider adding a sentence about opportunities for further exploration of animal contact (i.e. more robust consideration of household rodents and defining contact more precisely) in future studies, perhaps in the section related to limitations. But again, this is just a suggestion! I look forward to seeing the paper in print.

7. PLOS authors have the option to publish the peer review history of their article (what does this mean?). If published, this will include your full peer review and any attached files.

Reviewer #1: **Yes: **Claire Standley
